# Potassium Metabolism and Management in Patients with CKD

**DOI:** 10.3390/nu13061751

**Published:** 2021-05-21

**Authors:** Shinsuke Yamada, Masaaki Inaba

**Affiliations:** 1Department of Metabolism, Endocrinology, and Molecular Medicine, Osaka City University Graduate School of Medicine, 1-4-3, Asahi-machi, Abeno-ku, Osaka 545-8585, Japan; 2Kidney Center, Ohno Memorial Hospital, 1-26-10, Minami-Horie, Nishi-ku, Osaka 550-0015, Japan; inaba-m@med.osaka-cu.ac.jp

**Keywords:** potassium, potassium excretion, blood pressure, salt, hypertension, sodium, CKD

## Abstract

Potassium (K), the main cation inside cells, plays roles in maintaining cellular osmolarity and acid–base equilibrium, as well as nerve stimulation transmission, and regulation of cardiac and muscle functions. It has also recently been shown that K has an antihypertensive effect by promoting sodium excretion, while it is also attracting attention as an important component that can suppress hypertension associated with excessive sodium intake. Since most ingested K is excreted through the kidneys, decreased renal function is a major factor in increased serum levels, and target values for its intake according to the degree of renal dysfunction have been established. In older individuals with impaired renal function, not only hyperkalemia but also hypokalemia due to anorexia, K loss by dialysis, and effects of various drugs are likely to develop. Thus, it is necessary to pay attention to K management tailored to individual conditions. Since abnormalities in K metabolism can also cause lethal arrhythmia or sudden cardiac death, it is extremely important to monitor patients with a high risk of hyper- or hypokalemia and attempt to provide early and appropriate intervention.

## 1. Introduction

Abnormalities in potassium (K) metabolism are induced by a variety of factors. However, since K metabolism is regulated in a large part by the kidneys, most cases of hyper- and hypokalemia are caused by renal mechanisms [1]. Decreased renal function increases the risk of developing abnormal K metabolism, though aging of affected patients, and the increasing complexity introduced by various medications and dialysis treatments make the pathogenesis more complicated (Figure 1). In this article, the basics of K metabolism, the pathogenesis of abnormal K metabolism, and the relationships among factors related to K and its dynamics are examined, along with a review of relevant literature.

## 2. Distribution of K in the Body and Its Roles

Approximately 60% of adult body weight is water, two-thirds of which is intracellular and one-third extracellular. The major cation in intracellular fluid is K, while the major cation in extracellular fluid is sodium (Na). The total amount of K in the body is about 50–55 mEq/kg, about 98% of which is contained in intracellular (skeletal muscle, red blood cells, liver, etc.) and 1–2% in extracellular fluid. This concentration gradient (intracellular concentration: 150 mEq/L, extracellular concentration: 3.5–5.0 mEq/L) regulates excitatory conduction in nerve and muscle cells, as well as maintenance of osmotic pressure in body fluids and acid–base balance [2].

## 3. Regulatory Mechanisms of K in Kidneys

The normal daily intake of K in adults is 50–100 mEq, most of which is absorbed from the small intestine. Increased K in blood is taken up into cells by active transport through Na-K ATPase, with about 90% of excess K excreted in urine and about 10% in feces. When renal function is normal, serum K does not increase rapidly or significantly after K intake [3].

Of freely filtered K in the glomerulus, approximately 70% is reabsorbed in the proximal tubules and about 20% in the thick ascending limb (TAL) of Henle’s loop, with the remaining 10% regularly excreted (secreted) or reabsorbed by the cortical collecting duct (CCD). In the proximal tubules, reabsorption occurs by passive transport along with reabsorption of water and Na, and in the TAL by active transport by Na-K-2Cl cotransporters (NKCCs). In the CCD, several K channels, such as the renal outer medullary potassium channel (ROMK), Maxi-K, and Kv1.3, are expressed in the lumen and on both sides of blood vessels. K secretion is regulated by changes in the amount of K reaching the CCD lumen in response to changes in K concentration in blood, as well as the velocity of flow (urine volume) and negative potential in the lumen [4].

### 3.1. CCD Intraluminal Flow Velocity and Na Arrival Volume

When serum K concentration is increased, water and Na reabsorption become decreased due to increased K reabsorption in the proximal tubule and TAL, resulting in increases in flow velocity (urine volume) and Na arrival volume due to water diuresis in the CCD. This flow-dependent increase in K secretion is mediated by Maxi-K channels on the luminal side of CCD principal cells (PCs) and intercalated cells (ICs) [5]. When flow velocity in the lumen increases, transient receptor potential vanilloid (TRPV)–4 channels, which are also present in PC and IC cells, release stored intracellular calcium (Ca) and induce Ca influx from the lumen into the cells [6], thereby activating Maxi-K and enhancing K secretion [7]. When Na is reabsorbed in collecting tubules due to an increased amount of Na reaching the lumen, K secretion is enhanced by promotion of exchangeable excretion of K by the same cation.

### 3.2. ROMK and Maxi-K

A decrease in serum K concentration leads to decreased ROMK expression in PC cells of the CCD, endocytosis of ROMK protein, and degradation of channel protein from the luminal side membrane [8], as well as ROMK translocation into cells by activation of intracellular tyrosine kinase in the luminal side membrane of TAL, resulting in decreased K secretion [9]. Meanwhile, as K intake increases, Maxi-K expression on the luminal side of the collecting duct [10] and renal K excretion mediating Maxi-K [11] increase. In IC cells, K reabsorption by H-K ATPase activation in the luminal membrane is enhanced when serum K concentration is decreased. Although the existence of a K excretory function of IC cells during K ingestion has also been suggested [12], such a mechanism is not clear at present.

### 3.3. Aldosterone and Kallikrein

An elevated level of K in serum stimulates aldosterone secretion from the globular layer of the adrenal cortex, then aldosterone increases epithelial sodium channel (ENaC) and ROMK expression on the luminal side of PC cells, as well as Na-K ATPase on the vascular side, resulting in K secretion with Na reabsorption as the driving force [13]. In addition, hyperkalemia increases kallikrein production in junctional tubules, which enhances ENaC activity in PC cells, thereby promoting K secretion from ROMK and Maxi-K, and suppresses K reabsorption via H-K ATPase in IC cells.

### 3.4. Vasopressin (Arginine Vasopressin: AVP), Insulin, and Glucocorticoids

AVP and insulin enhance K secretion by increasing and activating ENaC, respectively [13]. While AVP also directly activates ROMK, K secretion is reduced during antidiuresis due to decreased flow velocity (urine volume); thus, the effects on K secretion are counterbalanced and suppressed. On the other hand, glucocorticoids appear to increase the glomerular filtration rate (GFR) and increase K secretion via increased CCD intraluminal flow velocity and Na arrival.

## 4. K Transportation in Intestinal Tract

Most ingested K is absorbed in the small intestine, with about 10% excreted in feces. There are two pathways for ion transport in intestinal epithelium; the intercellular collateral pathway, a passive transport pathway through the tight junction, and the transcellular pathway, an active transport pathway. K permeability tends to be greater in the upper small intestine (jejunum > ileum), and most of it is rapidly absorbed by the intercellular collateral pathway. The permeability of the intercellular collateral tracts of the colon is lower than that of the small intestine, though there are regulatory mechanisms in the colon for K absorption via H-K-ATPase in the cellular pathway and K secretion via ROMK on the lumen side. The presence of an enteric-derived factor that increases renal K secretion by K loading to the gastrointestinal tract has also been suggested [14].

In cases of VIPoma and colonic pseudo-obstruction, which cause marked watery diarrhea, K secretion into the stool is abnormally high, suggesting the involvement of vasoactive intestinal peptide (VIP) in intestinal K regulation [15]. Although Maxi-K is involved in K secretion in the colon [16], it has been reported that its expression in colonic cell epithelium and fecal K excretion is increased in patients with end-stage renal failure [17], thus indicating the existence of a compensatory mechanism for K excretion in the colon.

## 5. Metabolic Regulators of Intracellular K

### 5.1. Insulin

Since most ingested K is rapidly transferred from extracellular to intracellular fluid after absorption from the small intestine, the concentration of K in extracellular fluid is normally maintained in a range of 3.5–5.0 mEq/L without the appearance of hyperkalemia. Insulin promotes K uptake into skeletal muscle cells and hepatocytes by increasing Na-H exchange transport (NHE1) activity in the plasma membrane and also Na-K ATPase activity. Glucose is administered as glucose–insulin (GI) therapy for hyperkalemia to stimulate endogenous insulin secretion and prevent hypoglycemia. It has also been reported that oral glucose administration can increase endogenous insulin secretion and decrease plasma K concentration, even in hemodialysis patients with impaired renal function [18].

### 5.2. Catecholamine

Alpha-receptor stimulation inhibits intracellular K transport, while beta-receptor stimulation increases that by increasing intracellular cAMP, and activating protein kinase A and Na-K ATPase. In fact, β2-adrenoceptor agonists have been used for emergency treatment of hyperkalemia in chronic kidney disease (CKD) cases [19].

### 5.3. Intravascular pH

Under acidosis, H-transfer into cells is decreased, and thus the activity of NHE1 and intracellular Na concentration are decreased. As a result, Na-K ATPase activity and K transport into cells are reduced, resulting in an increase in serum K concentration. Generally, a decrease of 0.1 in intravascular (extracellular fluid) pH is thought to increase serum K concentration by approximately 0.6 mEq/L [20].

In cases of acidosis caused by accumulation of inorganic acids (e.g., HCl), hyperkalemia is exacerbated because K efflux from the cells is enhanced, whereas in acidosis caused by accumulation of organic acids (e.g., lactic acid), the concentration of K in serum remains nearly unchanged because organic acids enter the cells together with H. However, in severe cases of ketoacidosis and lactic acidosis, hyperkalemia is often observed due to the effects of insulin deficiency and hyperosmolarity. In patients with metabolic acidosis, a frequent complication of renal failure, the serum K concentration is more likely to increase as compared to those with respiratory acidosis. Thus, when renal function is impaired, treatment options that take into account regulation of intravascular pH are also necessary.

### 5.4. Osmotic Pressure

When plasma osmolality increases due to hyperglycemia, hypernatremia, or urea nitrogen accumulation, serum K concentration also increases, because the osmotic difference causes water to move out of cells, which increases intracellular K concentration and induces extracellular K transfer. In general, an increase in plasma osmolality of approximately 10 mOsm/kg is thought to increase serum K concentration in a range of 0.4–0.8 mEq/L. In particular, K control is likely to be difficult in diabetic kidney disease (DKD) patients with inadequate glycemic control; thus, strict glycemic control is important from the viewpoint of K management.

## 6. Epidemiological Results Showing Serum K Levels in Patients with CKD

An observational retrospective cohort study that used a Japanese hospital claims database (n = 1,022,087) reported that the prevalence of hyperkalemia was significantly higher in CKD patients (227.9; 95% confidence interval (CI): 224.3–231.5) as compared to all enrolled subjects (67.9; 95% CI: 67.1–68.8) (per 1000) [21]. On the other hand, in a study that examined the Japan Chronic Kidney Disease Database (J-CKD-DB) (n = 35,508), the prevalence of hyperkalemia in CKD stage G4 and G5 patients was found to be only 8.3% and 11.6%, respectively. However, though the serum potassium levels in stage G4 and G5 were significantly greater than those in G3 cases, there was little risk of rising above normal [(G3, G4, G5: 4.33 ± 0.44, 4.68 ± 0.73, 4.71 ± 0.76, respectively, (mean ± SD)] [22]. These findings support the existence of various compensatory mechanisms related to decreased renal potassium excretion.

## 7. Compensatory Mechanism of K Excretion in Renal Failure Patients

Even though urinary excretion of K decreases with a reduction in number of nephrons due to decreased renal function, K secretion per nephron and K excretion in the intestine are increased in a compensatory manner [23,24,25]. Furthermore, hyperkalemia requiring treatment is rare until the GFR is less than 10 mL/min [23,24]. Intestinal potassium transport mainly occurs by absorption in the jejunum and ileum, and secretion in the colon, though it has been reported that K excretion in the rectum is greater in hemodialysis and peritoneal dialysis patients compared to healthy subjects [26]. It is considered that elevated aldosterone in cases of renal failure increases ROMK in the colonic mucosa [17,27] and promotes K excretion via Na-K ATPase in the colonic mucosa [28]. Suppression of fecal K level by administration of spironolactone in patients with renal failure supports this speculation [29]. It should be noted that constipation and administration of a renin–angiotensin system inhibitor in patients with renal failure may contribute to refractory hyperkalemia and should thus be handled with caution. K is also excreted from the epidermis in the form of sweat, though to a lesser extent, while it has been reported that the concentration of K in sweat is higher in dialysis patients than in healthy subjects [30].

## 8. Recommended Daily Intake of K

The minimum daily requirement for K intake in adults, estimated based on its unavoidable loss through sweat, stool, urine, and other sources, is considered to be about 1600 mg (40 mEq). On the other hand, the WHO recommends a daily intake of 3510 mg (90 mEq) from the viewpoint of hypertension prevention [31,32]. Based on the above, the target amount of K in the *Dietary Reference Intakes for Japanese* is set at 2700–3000 mg per day, though actual intake is estimated to be around 2200–2400 mg (50–60 mEq).

Approximately 90% of K is excreted by the kidneys, and since healthy individuals can excrete more than 400 mEq per day, there is no upper limit regarding intake, as a normal diet will not result in excess K in the body. However, it takes several hours after ingestion for renal excretion to be completed, and K absorbed from the intestinal tract is first distributed extracellularly (in blood vessels). Thus, even oral intake by healthy individuals can cause transient hyperkalemia if it is rapid and in a large amount [33]. In this regard, cases of fatal arrhythmia due to supplements or salt substitutes containing large amounts of K have been reported [34].

## 9. Precautions for K Restriction in Elderly Patients with Renal Failure

Presently, Japan is a super-aged society, with more than 25% of the total population over the age of 65. The proportion of CKD patients in the elderly population is also high, with 30% of those over 70 and 40% of those over 80 years old meeting the definition of CKD [35]. While the importance of dietary restriction increases with progression of CKD, excessive restrictions may worsen the general condition of elderly individuals because they tend to have lower cognitive function and activities of daily living and are at higher risk of hyponutrition, frailty, and sarcopenia. Although target values for K restriction are the same for elderly and non-elderly patients, those who are elderly and have a small stature and low muscle mass are more likely to develop hyperkalemia because they generally take up less K intracellularly [36], while elderly patients with a low-K pool due to diuretics use tend to develop hypokalemia. In particular, management of K levels in elderly patients with renal failure must be individually tailored.

## 10. K Restriction Target Value

### 10.1. Target Value for Patients with Conservative Renal Failure

For CKD patients, it is recommended that the serum K level should be regulated in the range of 4.0–5.4 mEq/L. However, since renal K excretion decreases as renal function declines, target values for K restriction have been set according to the severity of CKD (CKD stage). In these patients, K restriction starts to become necessary after stage G3b [37] (or eGFR 40 mL/min/1.73 m^2^ or lower) [38], while for those in stage G1 or G2, the recommended intake is about the same as that of healthy individuals (2700–3000 mg/day). K-rich fruits and vegetables are also rich in vitamins and dietary fibers and have a protective effect against hypertension and renal disorders due to their alkalinizing effect on body fluids [39]. Therefore, some countries and regions recommend a daily intake of 4000 mg or more for healthy individuals as well as those at high risk of developing kidney disease [40].

Since the risk of developing hyperkalemia (serum K concentration ≥5.5 mEq/L) increases in proportion to decreased renal function in CKD stage G3 and above [37], it is recommended to limit K intake to 2000 mg/day or less in stage G3b and 1500 mg/day or less in stage G4 and G5 patients. It has been reported that hyperkalemia negatively affects not only life prognosis but also renal prognosis [41] and significantly increases the risk of transition to end-stage renal failure [42]. On the other hand, it has also been demonstrated that the risk of death from the same degree of hyperkalemia is reduced as the CKD stage progresses from G3 to G5 [40]. These findings suggest that chronic mild hyperkalemia is protective against cardiotoxicity caused by severe hyperkalemia and that a patient with even mildly impaired renal function requires careful attention in regard to fatal arrhythmia caused by hyperkalemia. In CKD patients without hyperkalemia (serum K concentration ≤5.4 mEq/L), higher levels of K intake, shown by urinary K excretion, indicate better mortality and renal prognosis [43,44]. Therefore, it is important to carefully monitor serum K concentration and encourage consumption of balanced amounts of vitamins and dietary fiber obtained from vegetables, fruits, and other foods.

Meat and fish also contain large amounts of K [45]; thus, reducing protein intake will reduce K intake. However, emaciation and malnutrition tend to become chronic conditions in elderly individuals [46], and thus excessive protein restriction may adversely affect life expectancy [47,48]. With development of medical technology, the number of CKD patients is increasing, though protein intake tends to decrease due to alterations in dietary habits associated with decreased renal function, especially in the elderly [49]. Undernutrition induces chronic inflammation and atherosclerosis caused by increased protein catabolism [50] and significantly worsens the prognosis of CKD patients [51]; thus, it has been suggested that protein intake of approximately 1.3 g/kg has little effect on renal prognosis [52]. In particular, adequate nutritional management for older patients is necessary based on a comprehensive assessment of individual conditions, risks, and adherence.

### 10.2. Target Value for Hemodialysis Patients

The K concentration in dialysate solutions commercially available in Japan ranges from 2.0 to 2.5 mEq/L, while 40–110 mEq is usually removed during a single dialysis session. As a result, K restriction has been relaxed compared to conservative CKD treatment, with the target K intake for dialysis patients set at less than 2000 mg/day. Nevertheless, potassium poisoning/sudden death still accounts for 2.7–4.7% of deaths among hemodialysis patients in Japan, with the highest serum K levels seen before the start of the week, because of the two-day gap between dialysis treatments [53], and the mortality rate showing a tendency to increase on weekends [54], confirming the difficulty of K control in dialysis patients. On the other hand, a study of elderly dialysis patients reported that the risk of mortality increases with lower levels of albumin, urea nitrogen, phosphate, and K [55]; thus, it is necessary to carefully monitor not only restrictions but also appropriate dietary intake, especially in older patients.

### 10.3. Target Value for Continuous Hemodialysis Patients

Increases in duration and frequency of hemodialysis have been shown to improve life expectancy [56], and the number of patients undergoing continuous dialysis is increasing. Continuous dialysis is commonly performed at home 5–7 days per week. Since there is no special dialysate used in those cases and dialysis is performed with ordinary dialysate with a K concentration of 2.0–2.5 mEq/L, there is a risk of hypokalemia, hypocalcemia, hypophosphatemia, and excessive alkalinization. At present, only about 0.2% of hemodialysis patients in Japan are receiving daily dialysis; thus, there is no clear target for K intake, and it is necessary to respond to individual conditions.

### 10.4. Target Value for Peritoneal Dialysis Patients

It is estimated that about 3% of Japanese patients with end-stage renal failure are treated with peritoneal dialysis. Removal of K by peritoneal dialysis is due to diffusion by a concentration gradient between the peritoneal capillaries and peritoneal dialysate. Commercially available peritoneal dialysate does not contain K [57], and since K is continuously removed on a daily basis, the need for K restriction is low unless there is a decline in residual renal or peritoneal function, with an intake of 2000–2500 mg/day recommended, about the same as that for healthy subjects. In these patients, attention should be paid to the appearance of hypokalemia, such as when dietary intake is inadequate.

## 11. Vegetables with Low K Content

Fruit intake should be minimized in CKD patients, while vegetables should be exposed to water or boiled down, and then the cooking water discarded to remove K. However, restriction of fruits and vegetables can contribute to intractable constipation due to fiber deficiency, with that associated with increased risk of chronic inflammation and mortality [58]. In addition, exposure of vegetables to water or boiling causes leaching and breakdown of minerals and water-soluble vitamins other than K. Above all, daily stress associated with dietary restrictions can significantly reduce the quality of life for these patients. Therefore, development of vegetables with low K content has recently been promoted. Since K is one of the elements essential for plant growth and an excessive decrease in the plant body causes growth disorders [59], low-K-content vegetables are cultivated by adjusting the amount of K fertilization in the nutrient solution during the growth process. Using this method, low-K edible parts have been achieved in leafy greens such as spinach, leaf lettuce, Chinese lettuce, komatsuna, and mesclun [60,61], as well as in fruit vegetables including tomatoes, melons, and strawberries [59,60,61,62,63]. It has also been reported that consumption of low-K-containing melons suppresses the increase in serum K concentration in dialysis patients before and after eating [64]. However, low-K vegetables cannot be cultivated in ordinary outdoor soil because it is necessary to exclude the effect of K contained in soil, which requires a plant factory that can control the cultivation environment, such as hydroponics. Since this requires a great deal of cost and labor, only lettuce, which is relatively easy to cultivate, is currently widely distributed. The effects of low-K-content vegetables on the human body and quality remain unclear and future developments are anticipated in this field.

## 12. Evaluation of K Kinetics Using Urinary K Measurement

### 12.1. Fractional Excretion (FE)

Approximately the same amount of K received as intake is reabsorbed, mainly in the small intestine, with about 90% of it then excreted in urine. Thus, the amount of K intake is nearly equal to the amount excreted in urine without taking into account unusual excretion or loss by defecation. In hypokalemia cases, urinary K of 5–25 mEq/day is generally considered to indicate inadequate K intake or extrarenal loss from the intestinal tract (diarrhea, vomiting, ileus, etc.), while more than 25 mEq/day suggests renal K loss (excessive K intake or increased aldosterone action). To accurately determine the amount of K excretion, it is necessary to collect urine throughout an entire day, though that has often been avoided in recent years due to the possibility of nosocomial infection caused by multidrug-resistant *Pseudomonas aeruginosa* (MDRP). Therefore, fractional excretion (FE), which is the ratio of solute clearance according to renal function without urine storage, is often used. The FE value is calculated as (FEK) (%) = (urine K concentration × serum Cr concentration)/(serum K concentration × urine Cr concentration) and used to determine how primary urine filtered by glomeruli is regulated during its passage through the tubules. The standard value is 15–20%. For example, if FEK is more than 10% under hypokalemia, it is considered to indicate sustained K excretion. However, in patients with a markedly reduced GFR, FEK tends to be overestimated beyond the margin of error, and this calculation should only be used in CKD cases up to stage G3. Furthermore, there are diurnal variations in urinary K excretion, with that in the early morning tending to be lower than during the day. This should be kept in mind when evaluating with spot urine so as to avoid over- or underestimation in clinical practice [65].

### 12.2. Transtubular K Gradient (TTKG)

Since the kidneys are capable of excreting more than 400 mEq/day of K, impaired urinary K excretion is always present in hyperkalemia cases, caused by decreased GFR, or insufficient aldosterone secretion and action. Moreover, renal K excretion is strongly influenced by Na concentration, a major regulator, thus urinary K secretion is enhanced in patients treated with thiazides or loop diuretics, for example, due to the presence of large amounts of Na in the lumen of the cortical collecting ducts. On the other hand, urinary K secretion is suppressed in renal failure patients who have been on a high sodium diet and are suddenly subjected to a strict salt restriction. The transtubular K gradient (TTKG) is used as an index of aldosterone action in the cortical collecting ducts because it is thought that most K excreted into the tubules is due to aldosterone action in the main cells of the cortical collecting ducts. TTKG is calculated as (urine K concentration/serum K concentration)/(urine osmolality/plasma osmolality) and generally decreased (<3) in hypokalemia and increased (>8) in hyperkalemia cases [66]. When TTKG is less than 2 in hypokalemia, it is presumed to indicate non-renal loss of K and when more than 2, renal loss (especially aldosterone action) is presumed, while TTKG less than 6 in spite of hyperkalemia leads to suspicion of adrenal insufficiency [67]. TTKG does not require urine storage as in FEK, though it is not possible to evaluate cases of extreme polyuria (hypotonic urine: urine osmolality < plasma osmolality) with no free water production or extreme dehydration (urinary Na concentration <25 mEq/L), in which free water production in the collecting duct cannot be accurately determined. TTKG was originally based on the assumption that the osmotic pressure ratio is equal to the solvent (free water) ratio, because solutes are neither reabsorbed nor secreted after the collecting duct segment [68]. However, a mechanism of recycling large amounts of urea in the collecting duct (reabsorbed urea circulating in the interstitium) was later found [69], disproving the assumption on which TTKG was based. Nevertheless, TTKG remains important for estimating renal K dynamics, though it is necessary to combine FEK, blood renin/aldosterone levels, blood gas findings, and other factors, including TTKG, to evaluate K dynamics based on measurement of urinary K.

## 13. Hyperkalemia

### 13.1. Causes

The normal range of serum K concentration is 3.5–5.0 mEq/L. Hyperkalemia is diagnosed at 5.0 mEq/L or higher, with therapeutic intervention required at 5.5 mEq/L or more. The causes can be broadly classified into pseudohyperkalemia, increased extracellular shift, increased extrarenal K load, and renal K retention. Pseudohyperkalemia and extracellular shift should be ruled out first, and renal K retention can be suspected if a large K load is ruled out. A comprehensive evaluation based on GFR, FEK, TTKG, blood gases, renin–aldosterone levels, and others similar should be performed. Pseudohyperkalemia is often caused by problems during blood collection (hemolysis, over-tightening of the tourniquet, excessive grasping) and, though rare, when thrombocytosis (>1,000,000/mL) or leukocytosis (>100,000/mL) has developed, K in blood cells is released, resulting in high levels of K in measurements. Increased extracellular shift can be seen in crush syndrome, increased catabolism, infection, and severe acidosis. Most chronic hyperkalemia is renal in origin and in the absence of renal failure can be considered a broad form of high-K distal-type acidosis (type IV RTA). Type IV RTA is caused by impaired ENaC due to aldosterone deficiency or insufficiency, which results in decreased excretion of K and H in the distal tubule and hyperkalemia, causing K to move intracellularly and H to move extracellularly, resulting in increased intracellular pH, which then inhibits ammonia production and decreases acid excretion [70]. TTKG can differentiate between decreased flow into the collecting duct and decreased K secretion at the same site, though the majority of cases are the latter and can be determined based on blood renin/aldosterone levels and blood gases. The most common cause is chronic renal failure, with hyperkalemia usually a problem in CKD stage G3b or later [37] (or eGFR 40 mL/min/1.73 m^2^ or less) [38]. On the other hand, in patients with cardiac or renal disease, hyperkalemia is often associated with medications that suppress the renin–angiotensin–aldosterone system, such as angiotensin-converting enzyme (ACE) inhibitors, angiotensin receptor blockers (ARBs), and K-retaining diuretics.

### 13.2. Symptoms

Severe and rapid hyperkalemia of 6.0 mEq/L or more (6.5–7.0 mEq/L or more in patients with end-stage renal failure) causes numbness and muscle weakness in the limbs, starting from the terminal muscles, and also arrhythmia, while 8.0 mEq/L or more results in bradycardia, dyspnea, and loss of consciousness. These symptoms are the result of depolarization of cell membranes due to increased K concentration in extracellular fluid, which enhances and sometimes attenuates excitatory cell functions of the heart, muscles, and nerves that depend on membrane potential. Electrocardiogram results show tent-like T waves, P wave flattening, QRS widening, bradycardia with junctional rhythm, and ventricular fibrillation. Since the rate of increase in serum K concentration is also involved in their appearance, ECG changes are often not seen in patients with end-stage renal failure or persistent hyperkalemia (Table 1).

### 13.3. Treatment

Because of the possibility of fatal arrhythmia in the acute phase, intravenous Ca (to stabilize myocardial cell membranes), insulin, beta 2 stimulants, bicarbonate (to promote intracellular K shift), fluids (to correct dehydration), diuretics, ion exchange resins, and emergency hemodialysis (to promote extracorporeal K excretion) should be administered in the best combination for each patient’s individual condition. Although hemodialysis is the most reliable method, it is necessary to re-measure serum K concentration 1–2 h after completion because even if the concentration becomes normalized by hemodialysis, K may be transferred from the intracellular pool again, resulting in hyperkalemia. Should persistent hyperkalemia be observed in chronic renal failure cases, diet and medications should be checked. K-containing salt substitutes in patients with a salt-restricted diet are easily missed and require attention. When K is difficult to control by dietary K restriction, consider the use of cation exchange resin preparations, such as sodium polystyrene sulfonate (PS-Na), Ca polystyrene sulfonate (PS-Ca), and sodium zirconium cyclosilicate (ZS-9), which chelate K in the intestinal tract and excrete it outside the body. PS is a polymeric adsorbent and may cause side effects, including constipation, abdominal pain, and fullness due to distention in the intestinal tract, and PS-Ca use is contraindicated in patients with an intestinal obstruction due to the risk of intestinal perforation. In salt-restricted patients, K-containing salt substitutes are easily missed and require attention. When control is difficult with dietary K restriction, use of cation exchange resin preparations, such as sodium polystyrene sulfonate (PS-Na), Ca polystyrene sulfonate (PS-Ca), and sodium zirconium cyclosilicate (ZS-9), which chelate K in the intestinal tract and excrete it outside the body, should be considered. PS-Ca is a polymeric adsorbent and may cause side effects including constipation, abdominal pain, and fullness due to distention in the intestinal tract, and its use is contraindicated in patients with an intestinal obstruction due to the risk of intestinal perforation. However, the recently developed ZS-9 has been shown to be associated with a lower risk of intestinal perforation as it is a non-polymeric, while PS-Na has about twice the K adsorption capacity of PS-Ca, though there is a risk of causing intestinal necrosis when used in combination with D-sorbitol, a sugar laxative [71,72,73]. PS is insoluble in water and when water is absorbed in the colon hard stools are produced, causing an increase in intestinal pressure. Furthermore, the decrease in intestinal blood flow induced by water removal by dialysis is thought to cause ischemic enteritis, which in turn results in intestinal perforation when it becomes severe. ZS-9 adsorbs monovalent K preferentially over divalent cations such as Ca and magnesium (Mg); thus, it is relatively efficient in excreting K [74]. Nevertheless, the maximum amount of K adsorption by these K-adsorbent resin preparations is approximately 1 mEq per gram of active ingredient and dietary therapy must be performed in parallel. It should also be noted that PS-Na is less effective when taken with a Ca preparation and that PS-Ca may contribute to hypercalcemia due to Ca release from the drug.

## 14. K Dynamics in Diabetic Dialysis Patients

While a function of insulin is translocating K into cells [75], in diabetes mellitus (DM) patients, such translocation tends to be inhibited due to decreased endogenous insulin secretion [76]. In addition, in cases of diabetic nephropathy, there is decreased renin activity and [77] metabolic acidosis due to ketone body accumulation.

We previously investigated the relationship between protein intake and K dynamics in 42 maintenance hemodialysis patients (22 DM, 20 non-DM). Their clinical characteristics are shown in Table 2. In the DM group, the normalized protein catabolism rate (n-PCR) from post-weekend hemodialysis to pre-Monday or pre-Tuesday hemodialysis was significantly and positively correlated with serum K level, as well as interdialytic serum K gain at the pre-hemodialysis examination (Figure 2a,b), whereas there was no such relationship in the non-DM group (Figure 2c,d). These results were similar among the patients, after excluding those using insulin and/or cation exchange resin products, and the relationships did not change after adjusting for age, gender, and hemodialysis duration (unpublished data). Our findings suggest that the defense mechanism against K elevation is weakened in T2DM hemodialysis patients compared to non-DM patients.

## 15. Hypokalemia

### 15.1. Causes

A serum K concentration of 3.5 mEq/L or less is used for diagnosis of hypokalemia, which is the second-most common electrolyte abnormality after abnormal Na concentration. Since 98% of K in the body exists in an intracellular location, with 70–80% in muscle, elderly individuals, especially those with low muscle mass, are prone to K deficiency due to decreased total K in the body. The causes of hypokalemia can be broadly classified into increased intracellular transport, decreased K uptake, and K loss (renal and extrarenal) [78], with renal loss the most frequent.

Increased intracellular transport can be caused by medications (e.g., insulin, beta-stimulants), familial hypokalemic periodic tetraplegia, and hyperthyroidism, as well as others, while renal loss is induced by medications (e.g., diuretics), increased renin–angiotensin system, and tubular dysfunction (Bartter syndrome, Gitelman syndrome, Liddle syndrome, etc.). As for extrarenal K loss, that can be caused by prolonged poor food intake, severe vomiting, and diarrhea. Since renal K excretion persists for approximately one week even after a decrease in serum K concentration, delayed renal adaptation to the disease in the acute phase may contribute to worsening of the disease.

Although hypokalemia is rare in patients with end-stage renal failure due to reduced K excretion, it can occur in individuals with reduced dietary intake because the concentration of K in standard hemodialysis fluid is set at 2–2.5 mEq/L and standard peritoneal dialysis fluid does not contain K. A survey by the Japanese Society of Dialysis Therapy found that hypokalemia can occur in patients with low dietary intake and 8% of patients had a post-dialysis K concentration of less than 3 mEq/L. In addition, the incidence of hypokalemia during CHDF is very high, ranging from 4% to 24% [79,80]. In CKD patients, not only hyperkalemia but also a serum K level below 3.5–4.0 mEq/L [41,81] are significant risk factors for total mortality, and there are several reports of sudden cardiac death of hemodialysis patients due to hypokalemia [82]. In particular, elderly patients should be carefully monitored to ensure that serum K levels are not too low.

### 15.2. Symptoms

Gastrointestinal symptoms such as vomiting and anorexia, muscular symptoms such as weakness and muscle weakness, impaired urine concentration (polydipsia, polyuria), and impaired insulin secretion (glucose intolerance) are observed when the serum K concentration is 2.5–3.0 mEq/L. When that concentration is lower than 2.5 mEq/L, tetraplegia, respiratory paralysis, ileus, and ventricular arrhythmia appear. The first change observed in ECG results is a decrease in the T wave, then ST depression and T waves become flat or negative as the K concentration decreases further, with U waves becoming apparent below 2.7 mEq/L, and extrasystoles, tachyarrhythmia, and 2–3 degrees of atrioventricular block noted at a concentration below 2.0 mEq/L [83], while torsades de pointes, ventricular fibrillation, and cardiac arrest may also occur. In hemodialysis patients, a sudden change in serum K level associated with dialysis can cause ventricular premature contractions and QT prolongation, thus leading to torsade de pointes and ventricular fibrillation.

Hypokalemia also leads to a variety of tubulointerstitial pathologic and functional changes. As for pathologic changes, those include vacuolar degeneration of the proximal tubules [84,85], interstitial infiltration of mononuclear cells [86], and renal cysts [87] (Figure 3). Chronic hypokalemia due to an eating disorder is a risk factor leading to end-stage renal failure [88]. It has also been reported that hypokalemia predisposes to acute kidney injury from medications such as gentamicin and amphotericin, while impaired NaCl reabsorption due to ROMK suppression in TAL has been speculated to be a factor contributing to impaired urine concentration [78].

### 15.3. Treatment

Hypokalemia associated with muscle paralysis and arrhythmia requires parenteral K replacement. Care should be taken to strictly adhere to the dose given (<60–80 mEq/day), K concentration in the infusion (peripheral vein: <40 mEq/L, central vein: <100 mEq/L), and administration rate (<20 mEq/hour) to avoid the risk of a rapid rise in concentration of K in serum. To avoid such an increase, K supplementation should be given orally, except in emergency cases. KCl is frequently used as an oral drug because it can also correct metabolic alkalosis. However, it is somewhat difficult to absorb and may cause ulcerations in the intestinal tract; thus, organic acid K salts (K aspartate, K gluconate), which are well absorbed and less likely to cause mucosal lesions, are often used. In RTA-induced hypokalemia, organic acid K salts should be used because of acidosis. Moreover, K-retaining diuretics are also effective for chronic hypokalemia, though care should be taken to avoid development of hyperkalemia in CKD patients using ACE, ARB, or NSAID.

## 16. Relationship between Mg and K

Following Ca, K, and Na, the most abundant cation in the body is Mg. More than 99% of total Mg in the body is distributed intracellularly and it is the second-most abundant electrolyte after K in cells. Hypomagnesemia is diagnosed when the serum Mg concentration is less than 1.5 mg/dL, and it is estimated that about 40% of hypokalemia patients also have hypomagnesemia. Since Mg has an ROMK-mediated inhibitory function toward K secretion, it is thought that K secretion is enhanced in an Mg-deficient state [89]. The decrease in urinary K excretion after Mg supplementation seen in hypokalemic patients being treated with thiazides [90] supports this speculation.

## 17. Relationship between NH_3_ and K

NH_3_ is produced not only in the liver but also in the proximal tubules of the kidneys in approximately the same amount as that produced in the liver. Nonvolatile acids produced by the metabolism of ingested proteins are excreted in urine as NH4, which helps to maintain the acid–base balance in the body [91]. In CKD patients, NH_3_ production is reduced due to a decrease in number of functional nephrons, resulting in decreased excretion of nonvolatile acids and metabolic acidosis [92]. Phosphate-dependent glutaminase (PDG) is the key enzyme for NH_3_ production, and its activity is increased by acidosis and hypokalemia. Therefore, hyperkalemia acts to suppress NH_3_ production and correction of hyperkalemia is important to correct acidosis associated with CKD (Figure 3).

## 18. Relationship between Na and K

NaCl is regulated by reabsorption via thiazide-sensitive Na-Cl cotransporters (NCCs) in the distal tubule, ENaC in the collecting duct, and pendrin. Under low-K conditions, NaCl reabsorption is enhanced by activation of NCC [93] and pendrin [94], whereas under high-K conditions, urinary Na excretion is enhanced by K influx into cells via K channels in distal tubules [95]. Therefore, K is thought to be a factor that can alleviate the effects of hypertension caused by excessive salt intake and its antihypertensive effect has been known for a long time [96]. Ca and Mg have been reported to have antihypertensive effects by promoting Na excretion and vasodilation, respectively [97], though their effects are not as significant as those of K [98].

Recently, the antihypertensive effect of K was made more apparent in results of a randomized controlled trial of hypertensive patients [99]. A diet rich in fruits and vegetables not only improves salt sensitivity related to high blood pressure [100] and lowers blood pressure [101,102], but also reduces the risk of renal failure and cardiovascular events [103]. Furthermore, in recent years it has been recommended that the optimal serum K concentration be set higher in patients with acute myocardial infarction, heart failure, or hypertension [104] (Figure 3). In CKD patients, it has been observed that blood pressure tends to be higher in those with a low serum K level [105], while it has been reported that a K-rich diet in patients with CKD stage G3 lowers blood pressure without changing serum K level [106]. Thus, it is necessary to constantly monitor for any hyperkalemia development in these patients, while ensuring that the serum K level is not excessively lowered.

The urinary Na/K ratio has been recommended as an indicator of dietary salt and K intake [107]. Both urinary Na and K tend to be low in the early morning, and the urinary Na/K ratio also tends to be low in the early morning due to the larger variation of K than Na [65]. On the other hand, the urinary Na/K ratio in 24 h urine and spot urine has been shown to be strongly correlated in normotensive [108] and hypertensive individuals [109], as well as early renal failure patients (CKD stage 1–3) [110]. However, this relationship is not seen in cases of advanced renal failure (CKD stage 4, 5) and it is difficult to evaluate using urinary Na/K ratio [110].

## 19. Drug-Induced K Abnormalities

### 19.1. Hyperkalemia

Elderly individuals are physiologically susceptible to hyperkalemia due to an age-related decline in renin–aldosterone secretion, and this risk is further amplified in those complicated by impaired renal function. Although ACEs and ARBs are being used with increasing frequency for cardio-renal protection, they may cause transient increases in serum Cr and K, especially in cases of renal failure and diabetic nephropathy, and treated patients should be carefully followed after starting administration [111]. K-retaining diuretics are antihypertensive agents that inhibit Na reabsorption and decrease K excretion by inhibiting ENaC in the collecting ducts, though they may sometimes cause fatal arrhythmia due to hyperkalemia. Among available K-retaining diuretics, spironolactone is contraindicated in anuria or acute renal failure cases, and eplerenone is contraindicated for diabetic nephropathy with albuminuria or proteinuria patients, and in those with creatinine clearance less than 30 mL/min. In addition, β-inhibitors and NSAIDs are prone to cause hyperkalemia by altering the distribution of K in and out of cells, and renin secretion, respectively [112].

### 19.2. Hypokalemia

The causes of drug-induced hypokalemia can be broadly divided into intracellular K shift, enhanced renal K excretion, and enhanced extrarenal K excretion. Beta-stimulants, glucose loading, insulin, xanthines, risperidone, and quetiapine promote intracellular K shift. Loop diuretics inhibit NKCC of the TAL, and thiazide diuretics inhibit NCC of the distal tubule, both of which increase K excretion by increasing the amount of Na reaching the collecting duct. High doses of penicillin, aminoglycoside, or platinum anticancer drugs stimulate K secretion by loading more nonabsorbable anions into the distal tubule. Fludrocortisone is a potent stimulator of Na reabsorption and K excretion, and hypokalemia may occur even at low doses. Most drugs that increase extrarenal K excretion, with the exception of cation exchange resins, are laxatives. In addition to direct K loss, secondary aldosteronism associated with extracellular fluid loss further enhances K excretion [113].

## 20. Conclusions

K is an electrolyte essential for maintenance of body functions, though its optimal concentration range is narrow, making it extremely vulnerable to metabolic abnormalities, especially when renal function is impaired. When managing K levels in CKD patients, it is extremely important to accurately determine and carefully monitor conditions in individual cases and attempt to appropriately intervene at an early stage.

## Figures and Tables

**Figure 1 nutrients-13-01751-f001:**
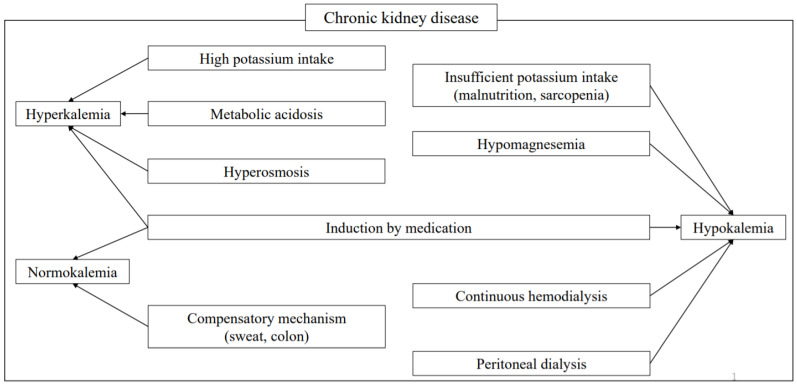
Factors affecting potassium metabolism in chronic kidney disease. Various factors cause abnormal K metabolism in affected patients. Although sweat and the intestinal tract provide compensatory mechanisms, correction with medication is also usually needed.

**Figure 2 nutrients-13-01751-f002:**
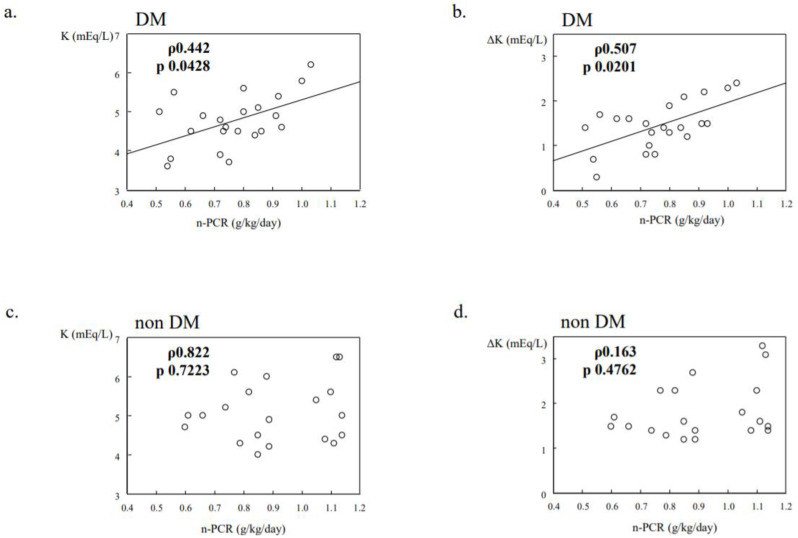
Potassium dynamics in DM/non-DM dialysis patients (data from previous research). N-PCR was significantly and positively correlated with serum K levels at the pre-Monday or pre-Tuesday HD session, and interdialytic serum K gain from post-weekend HD to the next session (**a**,**b**), whereas no relationship was noted in the non-DM group (**c**,**d**).

**Figure 3 nutrients-13-01751-f003:**
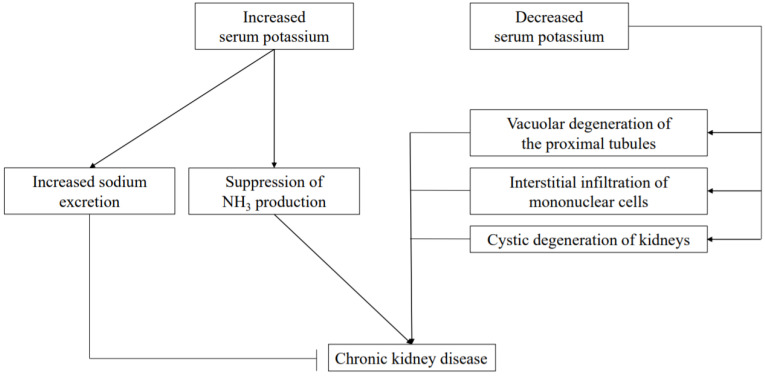
Role of potassium in CKD pathogenesis. Elevated serum K has protective effects on the kidneys by promoting salt excretion and reducing renal function by acidosis due to suppression of NH_3_ production. On the other hand, decreased serum K reduces renal function with a direct negative effect on renal tissue.

**Table 1 nutrients-13-01751-t001:** Various diseases/conditions associated with potassium deficiency and toxicity.

Deficiency	Toxicity
Gastrointestinal symptoms	Neuromuscular symptoms
vomiting/anorexia	lip numbness
ileus	muscle weakness
Neuromuscular symptoms	dyspnea due to respiratory paralysis
tetraplegia	Arrhythmia
muscle weakness	bradycardia
dyspnea due to respiratory paralysis	ventricular fibrillation
Impaired insulin secretion	ventricular flutter
Kidney disorders	cardiac arrest
impaired urine concentration	Lassitude
tubulointerstitial changes	Loss of consciousness
Arrhythmia	
extrasystoles	
tachyarrhythmias	
atrioventricular block	
ventricular fibrillation	
Lassitude	
Loss of consciousness	

**Table 2 nutrients-13-01751-t002:** Baseline clinical characteristics of non-DM and T2DM HD patients at the pre-Monday or pre-Tuesday HD session (data from previous research).

	All HD Patients (*n* = 42)	Non-DM Group (*n* = 20)	T2DM Group (*n* = 22)	*p* Value
Age, years	65.5 ± 11.1	66.1 ± 10.7	65.0 ± 11.7	0.6960
Gender, male/female	29/13	17/5	12/8	0.2322
BMI, kg/m^2^	22.7 ± 4.5	21.3 ± 4.3	24.0 ± 4.3	0.0494
HD duration, years	4.6 (2.8–6.4)	6.1 (3.4–8.0)	4.2 (2.1–5.5)	0.0365
Interdialytic BW gain, %	5.2 (4.3–5.8)	5.3 (4.5–6.9)	5.1 (4.3–5.5)	0.2733
Serum urea nitrogen, mg/dL	61.3 ± 15.3	65.0 ± 17.3	58.0 ± 12.8	0.1511
Cre, mg/dL	10.0 ± 2.5	10.3 ± 2.7	10.5 ± 2.5	0.7818
Alb, g/dL	3.7 (3.4–3.8)	3.5 (3.2–3.7)	3.7 (3.5–4.0)	0.0298
Casual plasma glucose, mg/dL	121.0 (101.0–149.0)	107.0 (93.0–127.5)	141.0 (117.0–162.0)	0.0048
Glycoalbumin, %	16.6 ± 3.0	14.9 ± 2.1	18.1 ± 3.0	0.0004
Na, mEq/L	139.8 ± 3.4	139.3 ± 4.5	140.2 ± 2.2	0.6935
K, mEq/L	4.9 ± 0.7	5.1 ± 0.8	4.8 ± 0.7	0.2775
Cl, mEq/L	105.9 ± 3.3	105.6 ± 4.1	106.2 ± 2.4	0.7039
PCR, g/day	45.72 (42.09–53.29)	45.65 (42.14–52.51)	46.38 (40.88–53.29)	>0.9999
n-PCR, g/kg/day	0.834 ± 0.181	0.911 ± 0.185	0.765 ± 0.149	0.0135
pH	7.34 (7.33–7.36)	7.34 (7.32–7.37)	7.34 (7.33–7.36)	0.5371
HCO_3_, mEq/L	19.8 ± 2.3	19.2 ± 2.7	20.4 ± 1.8	0.1658
AcAc, µmol/L	25.0 (21.0–44.0)	24.5 (19.0–36.5)	27.0 (22.0–53.0)	0.2360
β-HB, µmol/L	20.5 (15.0–40.0)	17.0 (12.5–33.0)	30.5 (19.0–64.0)	0.0070
AcAc /β-HB ratio, µmol/µmol	1.19 (0.75–1.47)	1.35 (1.07–1.79)	0.97 (0.69–1.24)	0.0136

Values in parentheses show range.

## Data Availability

Not applicable.

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
