# Peer review of "Potassium Metabolism and Management in Patients with CKD"

_nutrients, 2021, doi:10.3390/nu13061751_

Round 1

Reviewer 1 Report

This narrative review explains the role of potassium metabolism and management in CKD. The article is interesting, and it addresses a current topic, however, there are a few areas that need to be added/improved before publishing.

  • Please add an illustrative diagram to show the role of potassium in CKD pathogenesis.
  • Give a summary table of various diseases/conditions associated with potassium deficiency and toxicity in the kidney.
  • Give details of the physiology of potassium homeostasis including the role of the transporter.
  • How the presence of different metals will influence the potassium function. Please explain.
  • Please add a summarizing figure that supports the conclusions section.

Author Response

This narrative review explains the role of potassium metabolism and management in CKD. The article is interesting, and it addresses a current topic, however, there are a few areas that need to be added/improved before publishing.

The reviewer’s comments and suggestions are highly appreciated. The manuscript has been revised accordingly.

  • Please add an illustrative diagram to show the role of potassium in CKD pathogenesis.

As indicated, Figure 3 “Role of potassium in CKD pathogenesis” has been added to the revised version.

  • Give a summary table of various diseases/conditions associated with potassium deficiency and toxicity in the kidney.

Table 1 “Various diseases/conditions associated with potassium deficiency and toxicity” is now included in the revised version. As a result, original Table 1 has been renumbered as Table 2.

  • Give details of the physiology of potassium homeostasis including the role of the transporter.

The physiology of potassium homeostasis is indicated in Sections 3 (kidney transporter), 4 (intestinal tract), 5 (intracellular), and 6 (compensatory mechanism). Unfortunately, data regarding additional details are not available for this report.  

  • How the presence of different metals will influence the potassium function. Please explain.

Few reports regarding the association of potassium with metals other than magnesium have been presented. Thus, only the association with magnesium is mentioned in this review, in Section 15.

  • Please add a summarizing figure that supports the conclusions section.

As indicated, new Figure 1 “Factors affecting potassium metabolism in chronic kidney disease” has been added to the revised version. As a result, the original Figure 1 has been renumbered as Figure 2.

Reviewer 2 Report

The authors of this manuscript reviewed the findings related to potassium metabolism and management in CKD. I have several comments for this manuscript.

  1. Potassium management is related to the ingested potassium; however, authors tend to avoid making discussions on the methods for identifying the individual daily potassium intake and related nutrition epidemiological findings (e.g. gold standard method for identifying the individual daily potassium intake has been determined as 24-hour urine collection in the text books in nutrition epidemiology). Please add related statements in this review.

  1. Diurnal variation of urinary potassium concentration (also sodium concentration) have been reported previously (see PMID: 28123179). How does the potassium metabolism affect changes in urinary potassium concentration and resulted as a phenomenon of diurnal variation in urinary potassium concentration? Please add related statements in this review.

  1. In the section 17, authors demonstrate the findings of relationship between Na and K. Previous review suggested to consider use of the urinary Na/K ratio for practical dietary sodium reduction and dietary potassium increase instead of using urinary Na or K alone (see PMID: 28678188). In this previous review, new methods were introduced; urinary Na/K ratio using mean value of multiple spot urine for estimating 24-hour value under normotensive individuals (see PMID: 24718298), hypertensive individuals w/wo medication (see PMID: 26310187), chronic kidney disease patients (see PMID: 30443006) and multi-ethnic populations (see PMID: 28039381). According to the findings shown in PMID: 30443006, latter stage kidney disease patients (CKD stages 4-5) would not be suitable for applying this method but other individuals (CKD stages 1-3) remained suitable. Thus, the discussion on relationship between Na and K has to be different among CKD stages (stages 1-3 vs 4-5). Moreover, the interaction between sodium and potassium metabolisms may vary across CKD stages among patients with CKD. Therefore, the discussion which authors had made in the statements need to be differentiated by the disease stages. Please add speculations and related statements for the underlying scientific mechanisms differentiated by the CKD disease stages.

  1. Furthermore, diurnal variation of urinary sodium-to-potassium ratio has been reported previously (see PMID: 28123179). How can the interaction between sodium and potassium metabolisms work and how can you explain this phenomenon? Please add related statements in this review.

    5. Authors need to add appropriate citations to support their statements throughout the main text. Supportive evidences were relatively less and insufficient. 

Author Response

The authors of this manuscript reviewed the findings related to potassium metabolism and management in CKD. I have several comments for this manuscript.

The comments and suggestions from the reviewer were very helpful. The manuscript has been revised accordingly.

  • Potassium management is related to the ingested potassium; however, authors tend to avoid making discussions on the methods for identifying the individual daily potassium intake and related nutrition epidemiological findings (e.g. gold standard method for identifying the individual daily potassium intake has been determined as 24-hour urine collection in the text books in nutrition epidemiology). Please add related statements in this review.

As the reviewer noted, the gold standard for determining potassium dynamics is 24-hour urine collection analysis. However, due to the risk of becoming a hotbed for infectious disease, 24-hour urine collection tends not to be performed in general practice. Thus, evaluations by spot urine analysis are primarily referred to. This point is described in Section 11-1, line 7.

  • Diurnal variation of urinary potassium concentration (also sodium concentration) has been reported previously (see PMID: 28123179). How does the potassium metabolism affect changes in urinary potassium concentration and resulted as a phenomenon of diurnal variation in urinary potassium concentration? Please add related statements in this review.

As indicated, a description regarding diurnal variations of urinary potassium concentration has been added to the revised version (see Section 11-2, line 19).

  • In the section 17, authors demonstrate the findings of relationship between Na and K. Previous review suggested to consider use of the urinary Na/K ratio for practical dietary sodium reduction and dietary potassium increase instead of using urinary Na or K alone (see PMID: 28678188). In this previous review, new methods were introduced; urinary Na/K ratio using mean value of multiple spot urine for estimating 24-hour value under normotensive individuals (see PMID: 24718298), hypertensive individuals w/wo medication (see PMID: 26310187), chronic kidney disease patients (see PMID: 30443006) and multi-ethnic populations (see PMID: 28039381). According to the findings shown in PMID: 30443006, latter stage kidney disease patients (CKD stages 4-5) would not be suitable for applying this method but other individuals (CKD stages 1-3) remained suitable. Thus, the discussion on relationship between Na and K has to be different among CKD stages (stages 1-3 vs 4-5). Moreover, the interaction between sodium and potassium metabolisms may vary across CKD stages among patients with CKD. Therefore, the discussion which authors had made in the statements need to be differentiated by the disease stages. Please add speculations and related statements for the underlying scientific mechanisms differentiated by the CKD disease stages.

The useful information in these comments is much appreciated. Based on the suggestion, a description regarding assessment of urinary Na/K ratio has been added to the revised version (see Section 17, line 20).

  • Furthermore, diurnal variation of urinary sodium-to-potassium ratio has been reported previously (see PMID: 28123179). How can the interaction between sodium and potassium metabolisms work and how can you explain this phenomenon? Please add related statements in this review.

Accordingly, a description about diurnal variations of urinary Na/K ratio has been added to the revised version (see Section 17, line 20).

  • Authors need to add appropriate citations to support their statements throughout the main text. Supportive evidences were relatively less and insufficient. 

The reference numbers following number 59 were incorrect in the original manuscript, which has been corrected in the revised version. Thank you for pointing out these details. Newly cited literature is noted in red text in the revised References section.

Round 2

Reviewer 2 Report

The authors replied to my previous comments. Several points need to be clarified.

  1. The author recognized that the gold standard for determining potassium dynamics is 24-hour urine collection analysis. From daily practice perspective it is understandable that 24-hour urine collection is difficult to apply, maybe due to the risk of becoming a hotbed for infectious disease. However, for research purpose, the accuracy of the data is essential. If unreliable data is utilized, results will be biased and unreliable data (e.g. CVD risks estimates) will mislead many researchers (see PMID: 27248297 for sodium case). Therefore, the author’s reasoning and the statements were not scientifically reasonable and unacceptable. The authors can refer to estimates of 24-hour urine values by spot urine analysis; however, these are not accurate enough to be defined as gold standard (see PMID: 28678188).

  1. Please add statements explaining and/or speculating the mechanism of diurnal variation of urinary potassium concentration and Na/K ratio from author’s perspective. The authors seem to avoid clarifying these points.

  1. The statements made by the authors remain not supported by citations in the first and second pages (from Sections 1 to 3). Furthermore, cannot find any citation in sections 3-3, 3-4, first half of section 4, most of the part of section 5-1, latter part of 5-3, 5-4, 11-1, most of the part of 11-2, most of the part of 12-1, 12-2, 12-3, 14, 18 and 19. The source of Japanese recommendation level introduced in section 7 need to be clarified. Authors need to avoid unclear source of their statements and clarify their sources/citations to support their statements throughout the main text.

Author Response

The authors replied to my previous comments. Several points need to be clarified.

The comments and suggestions from the reviewer were very helpful to improve our study. Accordingly, the manuscript has been revised.

  1. The author recognized that the gold standard for determining potassium dynamics is 24-hour urine collection analysis. From daily practice perspective it is understandable that 24-hour urine collection is difficult to apply, maybe due to the risk of becoming a hotbed for infectious disease. However, for research purpose, the accuracy of the data is essential. If unreliable data is utilized, results will be biased and unreliable data (e.g. CVD risks estimates) will mislead many researchers (see PMID: 27248297 for sodium case). Therefore, the author’s reasoning and the statements were not scientifically reasonable and unacceptable. The authors can refer to estimates of 24-hour urine values by spot urine analysis; however, these are not accurate enough to be defined as gold standard (see PMID: 28678188).

We fully agree with the reviewer’s comments. Indeed, we also think that a reliable study will require analysis of 24-hour urine collection. On the other hand, this review is focused on and outlines how to evaluate potassium kinetics in general clinical settings. We also do not believe that a 24-hour urine value estimate from spot urine analysis can be defined as a gold standard. These points are noted in Section 11, beginning on line 7.

  1. Please add statements explaining and/or speculating the mechanism of diurnal variation of urinary potassium concentration and Na/K ratio from author’s perspective. The authors seem to avoid clarifying these points.

We have outlined the general theory of the salt excretion-promoting effect of potassium, especially its importance in CKD, in Section 17. Some uncertainties remain about evaluations based on Na/K ratio and we have provided as much as we know about these issues. Unfortunately, we are not able explain the mechanism related to the diurnal variation of urinary potassium concentration and Na/K ratio. We appreciate your understanding in this regard.

  1. The statements made by the authors remain not supported by citations in the first and second pages (from Sections 1 to 3). Furthermore, cannot find any citation in sections 3-3, 3-4, first half of section 4, most of the part of section 5-1, latter part of 5-3, 5-4, 11-1, most of the part of 11-2, most of the part of 12-1, 12-2, 12-3, 14, 18 and 19. The source of Japanese recommendation level introduced in section 7 need to be clarified. Authors need to avoid unclear source of their statements and clarify their sources/citations to support their statements throughout the main text.Because the number of citations will be very large. Because the number of references becomes so much

Since the number of references would become very large, we have not cited those at a textbook-level content in the manuscript. Regarding the Japanese recommendation pointed out by reviewer, since it has not been published, we describe that as follows in Section 7, beginning on line 7, in the revised version.

“Based on the above information, the target amount of K in the Dietary References Intakes for Japanese of the Ministry of Health, Labor and Welfare in Japan (2015) is set at 2700-3000 mg per day, though actual intake is estimated to be around 2200-2400 mg (50-60 mEq).